# CCDA: A Novel Method to Explore the Cross-Correlation in Dual-Attention for Multimodal Sentiment Analysis

**Peicheng Wang, Shuxian Liu * and Jinyan Chen**

School of Information Science and Engineering, Xinjiang University, Urumqi 830017, China; 107552101310@stu.xju.edu.cn (P.W.); 107552103773@stu.xju.edu.cn (J.C.)
* Correspondence: liushuxian@xju.edu.cn

**Abstract:** With the development of the Internet, the content that people share contains types of text, images, and videos, and utilizing these multimodal data for sentiment analysis has become an important area of research. Multimodal sentiment analysis aims to understand and perceive emotions or sentiments in different types of data. Currently, the realm of multimodal sentiment analysis faces various challenges, with a major emphasis on addressing two key issues: (1) inefficiency when modeling the intramodality and intermodality dynamics and (2) inability to effectively fuse multimodal features. In this paper, we propose the CCDA (cross-correlation in dual-attention) model, a novel method to explore dynamics between different modalities and fuse multimodal features efficiently. We capture dynamics at intra- and intermodal levels by using two types of attention mechanisms simultaneously. Meanwhile, the cross-correlation loss is introduced to capture the correlation between attention mechanisms. Moreover, the relevant coefficient is proposed to integrate multimodal features effectively. Extensive experiments were conducted on three publicly available datasets, CMU-MOSI, CMU-MOSEI, and CH-SIMS. The experimental results fully confirm the effectiveness of our proposed method, and, compared with the current optimal method (SOTA), our model shows obvious advantages in most of the key metrics, proving its better performance in multimodal sentiment analysis.

**Keywords:** multimodality; sentiment analysis; attention mechanism



## 1. Introduction

Multimodal sentiment analysis (MSA) is an important branch in the field of artificial intelligence. It aims to capture and understand human sentiment or emotion contained in text, speech, images, or other types of data, usually including positive, negative, neutral, or more specific emotional states such as joy, sadness, and anger [1]. In recent years, with the popularity of online social platforms, a large amount of multimodal data has emerged on the Internet. By analyzing data containing multiple modalities, computers can perceive human sentiment in the data [2]. Multimodal sentiment analysis has attracted widespread attention and it is widely applied in social media analysis [3,4], market research [5,6], and human–computer interaction [7,8].

In early studies on multimodal sentiment analysis, researchers have mainly used the following approaches to process multimodal data: The first one is early fusion, by concatenating different unimodal features and subsequently processing the features using different classifiers or models. For example, Morency et al. [9] used an HMM to process three unimodal features simultaneously. Poria et al. used CNN- [10] and LSTM-based [11] models to explore the contextual relationships between modalities. Zadeh et al. [12] used Multi-Attention Block(MAB) and Long-Short Term Hybrid Memory(LSTHM) to capture and store dynamics in multimodal features separately. Haohan et al [13]. used a Select Additive Learning based on CNN to improve the generalization performance of the model. The second method is late fusion, by training modality-specific classifiers for

each modality and then predicting sentiment according to the weight of the classifier's results. For example, Glodek et al. [14] used Kalman filters as combiners for decision-making. Cai et al. [15] first used several different CNNs and subsequently vectorized and fused the output of the features from each CNN. Alam et al. [16] used Sequential Minimal Optimization (SMO, a variant of SVM) with different kernel functions and fused their results in decision-making.

Although these two methods were relatively simple, when dealing with modal features, the model is unable to capture intra- and intermodality dynamics efficiently, which may lead to poor model performance. The researchers then combined the advantages of early and late fusion and proposed hybrid fusion. Poria et al. [17] used deep CNNs to extract features and fused multimodal features using MKL and determine the weights of textual modalities using a decision fusion approach in the final stage. Kumar et al. [18] used gating mechanisms to selectively learn cross-modal interaction information and used the results for sentiment prediction. Zhang et al. [19] used a multihead attention mechanism to extract semantic and sentiment analysis, then train multiple base classifiers and ultimately fuse the decisions of the base classifiers.

Word-level fusion fuses different modalities in a temporal step to obtain cross-modal correlations. For example, Zadeh et al. [20] proposed a memory fusion network (MFN), by simulating interactions within modalities and generalizing the temporal relationships between different modalities, the sequence is ultimately unified based on the relationships between unimodal word-level features. Subsequently, in [21], they proposed a Graph-Memory Fusion Network and performed word-level fusion by using a dynamic fusion graph. Paul et al. [22] proposed an LSTHM-based model to obtain cross-modal interactions by performing a multi-stage fusion of modalities features between each time step. Wang et al. [23] proposed a Recurrent Attended Variation Embedding Network (RAVEN), by modeling the fine-grained structure in word segments and transforming word representations based on nonverbal dynamic information.

Tensor fusion uses different tensor-based computation methods to allow different modalities to interact. Zadeh et al. [24] proposed Tensor Fusion Network (TFN), modal correlations are obtained by computing the outer product between the feature tensors. Zhun et al. [25] proposed Low-rank Multimodal Fusion (LMF) to solve the problem of excessive complexity in tensor computation. Barezi et al. [26] introduced a modality-specific deconstruction method in the model to reduce information redundancy. Liang et al. [27] proposed a regularization method to learn cross-time and cross-mode correlations in low-rank tensors. Tao et al. [28] correlated features at the same time step and further proposed a dual low-order multimodal fusion method. Jin et al. [29] used LSTM-based and tensor-based CNN networks to capture intra- and intermodal dynamic information encapsulated in asynchronous sequences.

In recent years, a number of attention-based approaches have emerged. Through the attention mechanism, the model can be made to acquire inter- and intramodal correlations more efficiently. Poria et al. [11] used attention units to capture dynamics across modalities. In [30–34], multihead and self-attention were used to perform cross-modal interactions, respectively, and perceive emotional information that is not within the modality. In addition, the researchers used other attention-based methods such as Gate Recursive Units (GRUs) [35,36] and Graph Convolutional Networks (GCNs) [37].

Nevertheless, there are still two main challenges in current multimodal sentiment analysis research. The first one is inefficiency in modeling the intramodality and intermodality dynamics. Multimodal sentiment analysis requires processing data from different modalities and correlating them to capture sentiment. It also needs to deal with sentiment dependencies within a single modality to help the model understand sentiment more accurately. The second one is the way in which different modal features are fused. Effective integration of features from different modalities can improve the accuracy and robustness of the model, which is crucial for the reliability of sentiment analysis in practical applications.

In this paper, we use a transformer-based approach to capture sentiment information and extract dynamics within and between modalities, and we introduce the relevant coefficient for the fusion of multimodal features. In addition, we propose a new cross-correlation loss function for investigating the correlations between different levels of attention mechanisms. Specifically, we obtain the intermodality dynamics between the global representation and unimodal representation by using the cross-attention mechanism, which is the component of the Transformer, so that they can strengthen themselves by learning about each other in this process. At the same time, we obtain the intramodality information by using the self-attention mechanism for three unimodal features, respectively. In addition, in our research, we hypothesized that there is some correlation between different levels of attention mechanisms, so we propose the cross-correlation loss to assess the interrelationship between cross-attention and self-attention. The contributions of this paper can be summarized as follows:

- We propose CCDA, a hierarchical model that studies intra- and intermodality correlations by using self-attention and cross-attention, respectively. Moreover, we introduce a new method to fuse multimodal features efficiently.
- We innovatively introduce a new cross-correlation loss function to study the correlation between different levels of attentional mechanisms in more depth. The objective function is minimized to cut down redundant information, which can help our model to better perceive sentiment information.
- Extensive experiments demonstrate the effectiveness of our proposed methodology. Our model achieves comparable results to the state-of-the-art (SOTA) approach in all evaluation metrics on the CMU-MOSI, CMU-MOSEI, and CH-SIMS datasets.

## 2. Related Works

Multimodal sentiment analysis aims to obtain sentiment information from different types of data. It provides additional sources of information for affective computing and enables computers to understand and perceive human sentiment more accurately [1–4]. A key challenge in this area is determining how to efficiently fuse data from different modalities so that the model can recognize sentiment precisely. This section presents related works on multimodal sentiment analysis, including early fusion, late fusion, hybrid fusion, word-level fusion, tensor-based fusion, attention-based methods (Table 1 provides a brief description of several of these methods), and other recent research approaches.

Early fusion combines all of the features from different modalities (text, audio, and visual) into a single feature vector, which is then used for sentiment prediction using a classification algorithm or model. Morency et al. input three unimodal features into the HMM model simultaneously [9]. Poria et al. proposed a method using CNN networks [10], by feeding unimodal features into a multikernel learning classifier. Following this, [11] proposed an LSTM-based model to deal with different unimodal features and explored the contextual relationships between modalities. Zadeh et al. [12] concatenated the multimodal features at each time step, used Multi-Attention Block to capture the dynamics between different modalities, and used a Long-short Term Hybrid Memory to store the dynamic information associated with each modality. Haohan et al. [13] proposed a Select Additive Learning based on CNN model (SAL-CNN) to improve the generalization performance of the model. The advantages of these approaches are that they can take into account the correlation between different modality features at the early stage. However, premature fusion of unimodal features can prevent the model from capturing information about the dynamics within the modalities, which can affect the model's ability to perform fine-grained classification.

In contrast to early fusion, late fusion employs independent classifiers separately for unimodal data and then fuses the outputs of each model to generate the final multimodal representation, or votes on the results of each model. Glodek et al. [14] used the Kalman filter as the combiner for temporally ordered classifier decisions. It is a linear dynamical system based on a Markov model which is well suited for real-time classifier fusion. Cai et al. [15] used text CNN, image CNN, and multi CNN to process unimodal features

and multimodal features, respectively; they used logistic regression as a classifier with the vectorized features in the penultimate layer of different CNNs. In [16], Alam et al. generated their classification models using Sequential Minimal Optimization(SMO, which is a variant of SVM) for each feature set, and different kernel functions were used for different feature sets. Finally, the results of classifiers for different feature sets were fused using decision fusion. While late fusion helped the model to better integrate semantic information. However, the model is not able to obtain the interactions between modalities during the training process, which would prevent the model from capturing cross-modal dynamic information. In addition, it is usually accompanied by a more complex model structure and a larger number of parameters.

**Table 1.** Related works in multimodal sentiment analysis.

| Method Type | Description | Advantages | Flaws |
|---|---|---|---|
| Early fusion | Combines all of the features from different modalities into a vector. | Realizes modal interactions at the early stage. | Time asynchrony and information redundancy. |
| Late fusion | Employs independent classifiers separately for each modality. | Helps model to better integrate semantic information. | Usually involves more complex model structures. |
| Hybrid fusion | Combines the advantages of early fusion and late fusion | Balance the model's complexity. | Inefficiencies arising from the limitations of the backbone network. |
| Word-level fusion | Fuses word representation in the temporal dimension. | Helps model to understand the intrinsic relation of multimodal data. | Insufficient generalization. |
| Tensor-based | Utilizes various tensor-based methods to integrate information from different modalities. | Integrate multimodal data effectively and address the complexity and noise issues. | Excessive computation and lack of interpretability. |
| Attention-based | Learns the semantic and relevant information using different attention mechanisms or Transformer. | More flexible and accurate in processing temporal information and capturing interactions between different modalities. | Correlations between different attention mechanisms cannot be captured. |

Hybrid fusion combines the advantages of early fusion and late fusion, capitalizing on their strengths and compensating for their weaknesses, respectively. Poria et al. [17] proposed a method for extracting text features using deep CNNs and fusing multimodal heterogeneous features using MKL, in addition to a decision-level fusion method that determines the weights of the text modalities by the coupling of the sentiment modalities. Kumar et al. [18] used gating mechanisms to selectively learn cross-modal interaction information and utilized post-interaction results for sentiment prediction. Zhang et al. [19] used multihead attention to extract accurate semantic and affective information in the representation fusion stage, followed by training multiple base classifiers to make independent judgments on different unimodal representations in the decision fusion stage, and finally fusing base classifiers' decisions. The core idea of this approach is to allow features to be fused at different stages of the model while avoiding some of the potential problems of early fusion and late fusion. However, the limitations of the baseline model itself at that time made this type of fusion method not perform well enough.

Word-level fusion is a method that fuses word representations in the temporal dimension to capture the interrelationships between different modalities. This approach emphasizes word-level information interactions and helps to understand the intrinsic structure and semantic relatedness of multimodal data in more detail. In [20], Zadeh et al. proposed a Memory Fusion Network (MFN); they first modeled interactions within modalities and generalized temporal relationships across modalities, ultimately unifying sequences based on relationships between unimodal word-level features. Subsequently, in [21], they used a Graph-Memory Fusion Network to perform unimodal, bimodal, and trimodal word-level fusion for unimodal features, and captured intermodal interactions by using a dynamic

fusion graph. Paul et al. [22] proposed an LSTHM-based model, obtaining cross-modal interactions by performing multiple stage fusion of modalities features between each time step. Wang et al. [23] proposed Recurrent Attended Variation Embedding Network (RAVEN) by modeling the fine-grained structure in word segments and transforming word representations based on nonverbal dynamic information. Word-level fusion enables the integration of affective information from different modalities in word representations. However, this approach may result in the loss of specific affective information in the original modality, and the complexity of word-level fusion increases further when multiple different modalities are involved.

Tensor fusion utilizes various tensor-based methods to integrate information from different modalities. These methods can effectively integrate multimodal data and address the complexity and noise issues in the data. The tensor fusion network (TFN) [24] obtains the dynamic correlation between modes by calculating the outer product of bimodal and trimodal features. Zhun et al. [25] proposed a Low-rank Multimodal Fusion (LMF) method to solve the problem of excessive computational complexity in TFN, and utilized modality-specific low-rank factors for multimodal fusion to improve the efficiency. The Modality-based Redundancy Reduction Fusion (MRRF) [26] introduces a modal-specific decomposition method into the model, which removes redundant information from the dependency structure and leads to fewer parameters with minimal loss of information. Liang et al. [27] proposed a regularization method to minimize the rank of the tensor and learn correlations across time and modes in low-rank tensors. Tao et al. [28] correlated the features of a single time step between multiple modalities and further proposed a dual low-order multimodal fusion method to reduce computational complexity. Jin et al. [29] used LSTM-based and tensor-based CNN networks to discover intra- and intermodal dynamics, and encapsulated them in an asynchronous sequence. However, tensor fusion is often accompanied by high-dimensional data representations, which, again, increases computational complexity while causing data sparsity. On the other hand, tensor fusion reduces the interpretability of the model, which may limit the credibility and acceptance of the model in practical applications.

Attention mechanism (Especially Transformer [38], proposed by Google in 2017) plays a significant role in multimodal sentiment analysis; it helps models better understand and leverage the interconnections and semantic information between different modalities, and be more flexible and accurate in processing multimodal data. Chen et al. [39] and Poria et al. [11] used an LSTM-based model as well as attentional units to capture the dynamics across modalities. In [30–34], multihead and self-attention were used to capture relevant information within or across modalities. In addition, the researchers additionally used other methods, e.g., Gate Recurrent Unit (GRU) [35,36] and Graph Convolutional Network (GCN) [37]. The Transformer exhibits strong generalization capabilities, making it suitable for different types of multimodal sentiment analysis tasks.

In addition, there are other methods in multimodal sentiment analysis, such as multi-task contrastive learning [40], dynamic filtering mechanism [41], bidirectional multimodal dynamic routing mechanism [42], cross-modal hierarchical graph contrastive learning strategy [43], supervised contrastive learning [20,44], dynamic refined sentiment words [45], etc.

Previous studies have viewed modality self-attention and cross-modal attention as two separate units that cannot interact with each other. Therefore, in this study, we proposed Cross-Correlation in Dual-Attention model (CCDA) to capture the correlations that exist between the different attention layers, so that, after acquiring intra- and intermodal information, respectively, the model can also enable them to exchange information that is helpful for their respective learning. In addition, in the feature fusion stage, we propose a strategy to help the model converge quickly, by calculating the relevant coefficients between the unimodal self-attention features and the source feature representations to guide the multimodal feature fusion.

## 3. Methodology

### 3.1. Problem Definition

Multimodal sentiment analysis is a task that utilizes multiple modalities for the study of human sentiment. Typically, it includes three modalities: text, speech, and images. We define three modality feature sequences, $X_m = \{x_{m,1}, x_{m,2}, \ldots x_{m,n}\}$, and sample labels $Y = \{y_1, y_2, \ldots y_n\}$, where the modality is represented as $m \in \{t, a, v\}$ ($t$ stands for text, $a$ stands for audio, and $v$ stands for visual) and $n$ represents the number of samples in the dataset. Our goal is to input modality features $X_m \in \mathbb{R}^{T_m \times d_m \times n}$ into a model to obtain an accurate sentiment prediction label $y \in \mathbb{R}^1$, where $T_m$ and $d_m$ represent the sequence length and the dimension of modality features separately.

### 3.2. Model Structure

In this section, we provide a detailed overview of the architecture of the CCDA (Cross-Correlation in Dual-Attention) model, as shown in Figure 1. We first use three unimodal encoders to obtain the utterance representation $U_m^{d_m \times n}$ and embedding $F_m^{T_m \times d_m \times n}$ by using feature sequences $X_m$ for each modality separately, which $m \in \{t, a, v\}$, $U_m^{d_m \times n}$ originate from the feature representation in each unimodal encoder. This helps the model understand the semantic and sentiment information in each modality.

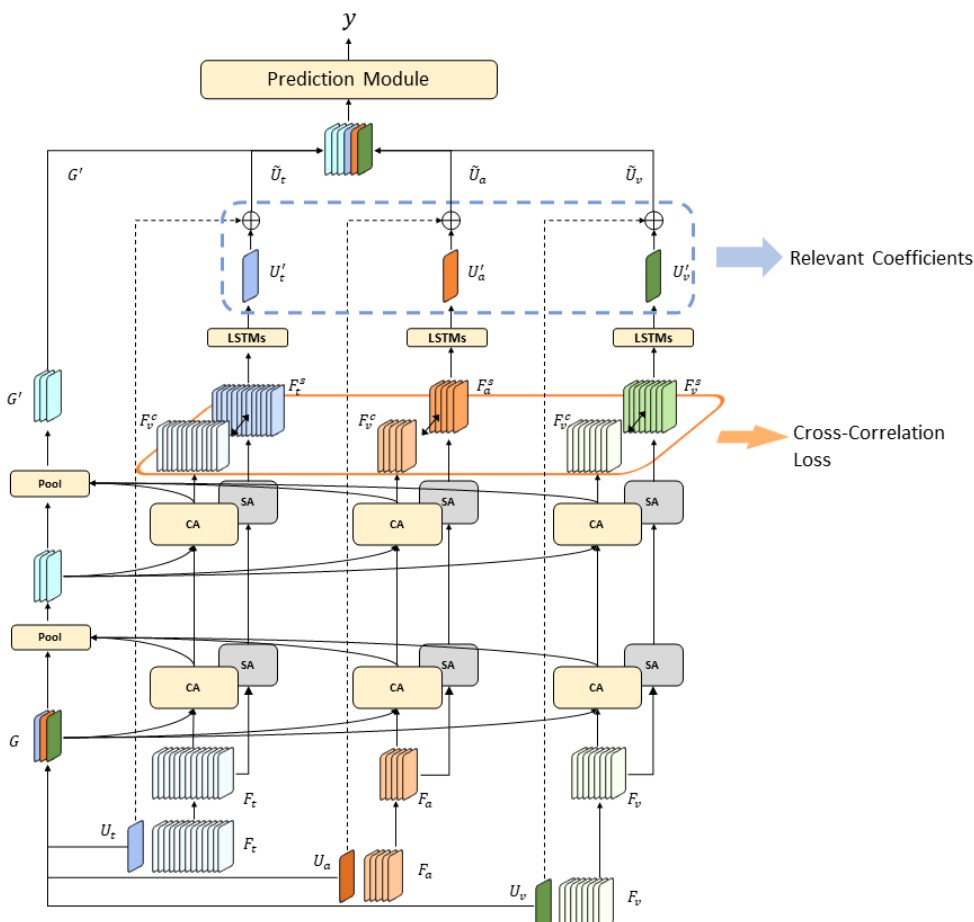

**Figure 1.** The structure of CCDA. The global representation $G$ consists of three unimodal representations $\{U_t, U_a, U_v\}$. The model processes the global representation $G$ and the unimodal features $F_m$ using the dual-attention to obtain new global and unimodal representations $\{G', \tilde{U}_t, \tilde{U}_a, \tilde{U}_v\}$ and fuses these representations for sentiment prediction. The unimodal features $\{F_t^S, F_a^S, F_v^S, F_t^C, F_a^C, F_v^C\}$ generated during this process are used to learn the correlation between the two attention mechanisms. The final objective function consists of the MSE loss $\mathcal{L}_{MSE}$ and the cross-correlation loss $\mathcal{L}_c$.

Next, we delve into the dual-attention mechanism (which contains self-attention and cross-attention), a core component of CCDA. By utilizing self-attention and cross-attention, CCDA can capture sentiment information and dynamics within a single modality (intramodality) and across different modalities (intermodality), respectively. This dual-attention mechanism enables the model to comprehensively analyze multimodal data and sentiment information, thereby improving the accuracy of sentiment analysis.

Following that, CCDA calculates cross-correlation losses between the embeddings generated by the two attention mechanisms while obtaining information about the intramodality and intermodality dynamics. This contributes to the indirect interaction between the two attention mechanisms and, thus, improves the model's performance. CCDA then uses relevant coefficients strategy to fuse the unimodal and multimodal representations obtained from these two attention mechanisms to generate the final sentiment representation.

In the following parts, we elaborate on the three main components of CCDA: unimodal encoders (Section 3.2.1), dual-level attention (Section 3.2.2), and fusion and prediction units (Section 3.2.3).

### 3.2.1. Unimodal Encoders

Similar to EMT [33], we employ the pretrained BERT model to encode textual tokens into context-aware word embeddings. Specifically, we notice that the [CLS] token of the BERT model contains a sequential representation of the text modality. Therefore, we use this token as the utterance representation for the text sequence, denoted as $u_t \in \mathbb{R}^{d_t}$. For the audio and visual modalities, we use LSTM recurrent neural networks to extract temporal information from the feature sequences. Ultimately, we select the hidden state of the last time step of the LSTM network for both the audio and visual modalities as their respective utterance representations: $u_a \in \mathbb{R}^{d_a}$ and $u_v \in \mathbb{R}^{d_v}$. Simultaneously, we need to process other tokens output by the BERT model and hidden states from LSTMs at different time steps for later use in self-attention and cross-attention mechanisms. These representations are denoted as $F_m \in \mathbb{R}^{T_m \times d_m}, m \in \{t, a, v\}$, representing the text, audio, and visual modalities, respectively.

$$
\begin{aligned}
F_t &= BERT(X_t) \\
F_a &= LSTM(X_a) \\
F_v &= LSTM(X_v)
\end{aligned}
\tag{1}
$$

### 3.2.2. Dual-Level Attention

Attention mechanisms help the model better understand multimodal sentiment data and perceive emotional information. They enable the model to capture dynamics within a single modality or between different modalities during the multimodal sentiment processing. The Transformer [38] is a language model in the field of natural language processing; it is based on dot-product self-attention mechanisms. It employs self-attention to infuse global semantic information and consider long-range dependencies for every word in the sequence. Furthermore, the multihead mechanism allows the model to learn different subspaces of semantics.

In simple terms, the Transformer processes the input sequence $H \in \mathbb{R}^{T \times d}$ with positional encoding; it defines Query as $Q = HW_Q$, Key as $K = HW_K$, and Value as $V = HW_V$, where $W$ represents the weight matrices during the feature sequence mapping process. Therefore, self-attention can be represented by Equation (2):

$$
Self\text{-}Attention(H) = softmax(\frac{QK^T}{\sqrt{d_k}})V
\tag{2}
$$

In MulT [30], the Query and K–V pair in the self-attention computation process come from different modalities. Thus, MulT captures the interaction between the two modalities. MulT combines three modality pairs and calculates bidirectional modality interactions for

each pair. As shown in Equation (3), for two modality feature sequences $H_1$ and $H_2$, MulT defines Query as $Q_1 = H_1 W_Q$, Key as $K_2 = H_2 W_K$, and Value as $V_2 = H_2 W_V$. It calculates cross-modal attention in two directions between a pair of modalities:

$$Cross\text{-}Attention(H_1 \rightarrow H_2) = softmax(\frac{Q_1 K_2^T}{\sqrt{d_k}})V_2$$

$$Cross\text{-}Attention(H_2 \rightarrow H_1) = softmax(\frac{Q_2 K_1^T}{\sqrt{d_k}})V_1 \tag{3}$$

EMT [33] concatenates three unimodal utterance representations into a multimodal global representation. Inspired by EMT [33], we concatenate the utterance representations from each modality $u_m$ as the global representation $G = Concat(u_t, u_a, u_v)$ during the cross-attention stage, where $m \in (t, a, v)$. Subsequently, we utilize a Transformer to calculate intermodality information between the modality feature sequences $F_m \in \mathbb{R}^{len \times d}$ and the global representation $G \in \mathbb{R}^{3 \times d}$, as shown in Figure 2 and Equation (4).

$$Attention(G \rightarrow F_m) = Cross\text{-}Attention(G \rightarrow F_m)$$

$$Attention(F_m \rightarrow G) = Cross\text{-}Attention(F_m \rightarrow G) \tag{4}$$

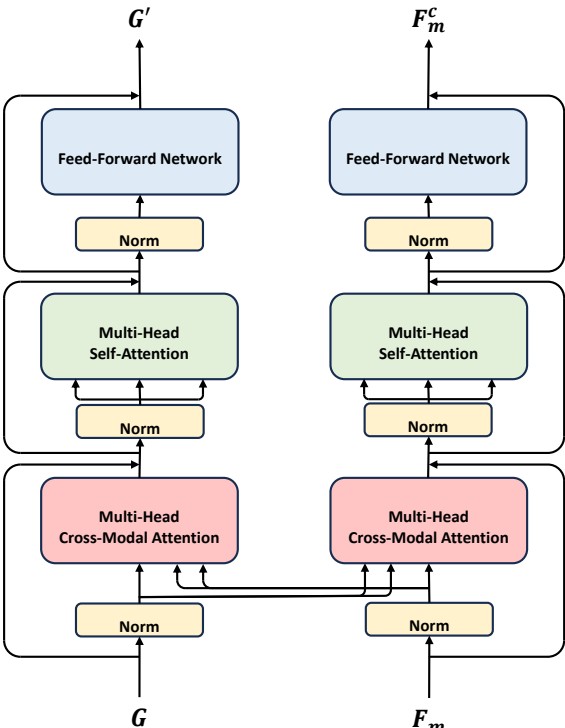

**Figure 2.** The structure of cross-attention. Cross-attention is used to capture dynamics between the global representation $G$ and unimodal representations $F_m$.

On the other hand, we utilize modality-specific Transformer encoder layers, denoted as $L_s$, to capture intramodality information for each modality individually (using Equation (2)). After encoding each modality, we use the self-attention mechanism in Transformer to process the unimodal feature sequences separately, in which the embedding at each position is able to learn the semantic and emotional information contained in the sequences.

MulT [30] used directional encoders for bimodal interactions separately, and subsequently augmented these dynamics with self-attention mechanisms. EMT [33] achieved cross-modal interactions by making global representations and unimodal sequences learn

from each other, while ignoring modality-specific information present in the self-attention unit. CCDA used both cross-modal attention and self-attention; first the two attention mechanisms were isolated, and then it used the cross-correlation loss to make them to interact after sufficiently learning the relevant intra- and intermodal information, respectively. This preserves the specificity information of the different attention mechanisms and optimizes the global representation by backpropagating the cross-modal feature sequences during the training progresses. After the feature sequences in the self-attention module learn the intermodal information of the cross-modal feature sequences, they are able to increase the perceptual field of the final multimodal features and increase the generalization performance of the model.

The use of dual-attention allows the model to process and analyze multimodal data at two different levels, intermodality and intramodality, for a more comprehensive understanding and interpretation of multimodal sentiment data.

### 3.2.3. Modality Fusion

After passing through the cross-attention stage, the model obtains intermodality information, which is reflected through the global representation $G'$, while in the self-attention stage, to maintain consistency with the global representation, we employ Bi-LSTMs to process the three single-modal feature sequences individually, obtaining each unimodal representation. Meanwhile, we propose the relevant coefficients, which are computed based on the relationship between the modal representation and the initial representation. Relevant coefficients strategy can fuse the representations obtained from dual-attention mechanisms and generate the final multimodal sentiment representation.

To be more specific, after learning intramodality information in the self-attention stage, the model utilizes Bi-LSTMs to transform unimodal feature sequences into feature representations $U'_m \in \mathbb{R}^{b \times d}$, which are specific to each modality. Subsequently, we calculate relevant coefficients based on the correlation between this representation and the initial modal representations $U_m \in \mathbb{R}^{b \times d}$:

$$r_m = \sum (Diag(tanh(U'_m) \bigotimes tanh(U_m)) - 1)^2 \tag{5}$$

where $\otimes$ denotes matrix multiplication, and $Diag(\cdot)$ represents all the diagonal elements of a square matrix. After obtaining the relevance coefficient $r_m$ for each modality, we multiply it with $U'_m$ to obtain the single-modal representation:

$$\widetilde{U}_m = r_m \times U'_m \tag{6}$$

Here, $r_m$ is the relevance coefficient specific to each modality, and $U'_m$ represents the feature representation of the corresponding modality obtained through Bi-LSTMs.

After obtaining the representations for both intermodality and intramodality $\{G', \widetilde{U}_l, \widetilde{U}_a, \widetilde{U}_v\}$, we concatenate the unimodal representations $\{\widetilde{U}_l, \widetilde{U}_a, \widetilde{U}_v\}$ with the global representation $G'$ to create the representation for the sample. Finally, we employ several linear layers in combination with activation functions to make predictions for the ultimate result.

$$y = Pred(Concat(G', \widetilde{U}_l, \widetilde{U}_a, \widetilde{U}_v)) \tag{7}$$

### 3.3. Cross-Correlation Loss

Most of the current research uses attention mechanisms to capture relevant information from both intramodality and intermodality, but few scholars consider the relationship between these two different attention levels. In order to extract this relationship in dual-attention, we propose a cross-correlation loss to obtain relevant information. By adding it to the objective function, the model is able to accomplish an undirected interaction between two different kinds of attention.

As shown in Figure 3, we use linear projectors to expand the feature sequence dimensions of the two different attention mechanisms and perform modality-specific matrix multiplication to obtain a set of matrices with a shape of (batch, length, length).

$$C_m = F_m^S \bigotimes F_m^C \tag{8}$$

where $C_m$ represents the cross-correlation matrix of the $m$ modality's feature sequences in two different attention mechanisms, $m \in \{t, a, v\}$. The diagonal elements in this matrix represent the correlation between the corresponding positions of the two feature sequences, while the off-diagonal elements represent the redundant information.

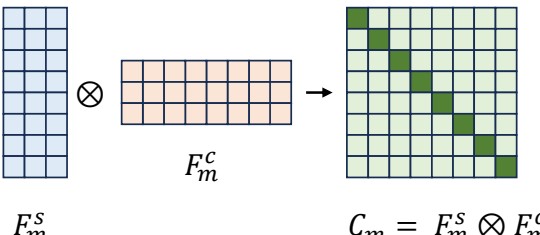

**Figure 3.** The cross-correlation matrix in dual-attention. We perform modality-specific matrix multiplication on the two types of unimodal feature sequences to obtain a cross-correlation matrix, and we use the diagonal elements of the matrix to represent the indirect interaction between these two feature sequences. The deeper the diagonal elements in the matrix $C_m$, the stronger the correlation between the two unimodal feature sequences at the corresponding positions is represented.

Taking the textual modalities of the samples in the CMU-MOSI dataset as an example, as shown in Figure 4, the model maximizes the diagonal elements in the intercorrelation matrix in order to capture the correlation between the different attentional mechanisms during the training process. At the same time, nondiagonal elements are minimized in order to reduce redundant information in this process.

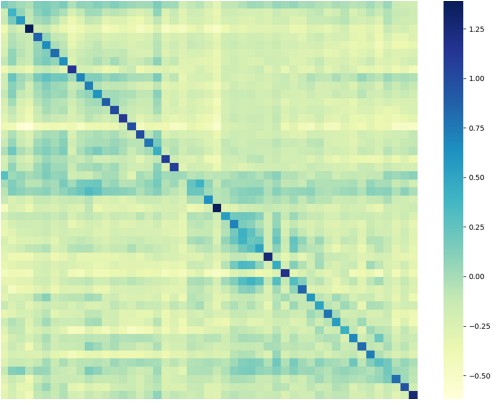

**Figure 4.** The cross-correlation matrix.

$$\mathcal{L}_{Corr} = \frac{1}{M} \cdot \sum_m^M (\sum_{i=j}^n (c_{ij} - 1)^2 + \sum_{i \neq j}^n c_{ij}^2) \tag{9}$$

As shown in Equation (9). The term $\sum_{i=j}^n (c_{ij} - 1)^2$ in $\mathcal{L}_{Corr}$ is the correlation term, which denotes the correlation between the sequence of modality features of $m$ in different attention mechanisms, and the other term $\sum_{i \neq j}^n c_{ij}^2$ is the redundancy term. Intuitively, the model increases the correlation between different attentional mechanisms by making the

diagonal elements of the cross-correlation matrix close to 1. At the same time, it reduces the redundancy term by making the off-diagonal elements of the cross-correlation matrix close to 0.

### 3.4. Loss Function

We use MAE and Cross-Correlation loss as the final objective function. As shown in Equations (10) and (11):

$$\mathcal{L}_{MSE} = \frac{1}{N} \cdot \sum_{i=1}^{N} |y_i - \hat{y}_i| \tag{10}$$

$$\mathcal{L} = L_{MSE} + \lambda \cdot L_{Corr} \tag{11}$$

Where $y$ denotes the true label of the sample and $\hat{y}$ denotes the predicted label of the model. Since the cross-correlation loss is calculated for all elements in the cross-correlation matrix, setting the weight of the cross-correlation loss too high in the objective function can cause the two attention mechanisms to lose their specificity and, thus, reduce the model performance. Therefore, we set a scaling factor $\lambda$ in the cross-correlation loss according to the expansion of the feature sequence dimension. We conducted ablation experiments on different scaling weights on two datasets, as shown in Section 4.3.

## 4. Experiment

### 4.1. Preparations

#### 4.1.1. Datasets

A multimodal dataset collects information from different modalities, such as text, speech, and vision, providing researchers with opportunities to gain a deeper understanding and analysis of sentiment expression. Three publicly available datasets are used in this article, including CMU-MOSI, CMU-MOSEI, and CH-SIMS. Figure 5 illustrates some samples from the CMU-MOSI and CMU-MOSEI datasets, and Figure 6 illustrates the CH-SIMS dataset.

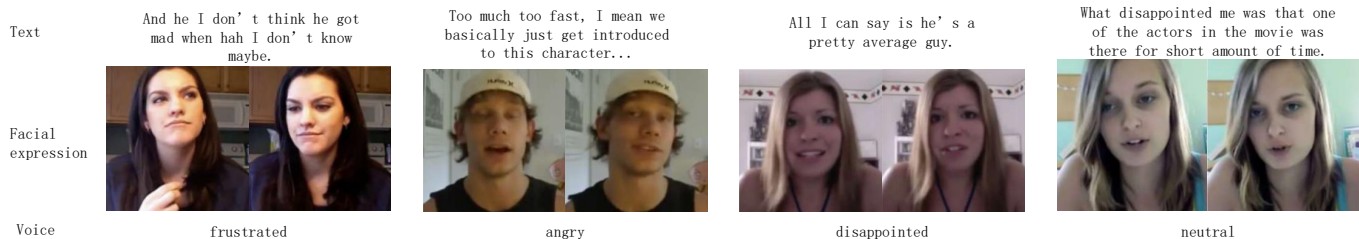

**Figure 5.** Examples in the CMU-MOSI and CMU-MOSEI datasets.

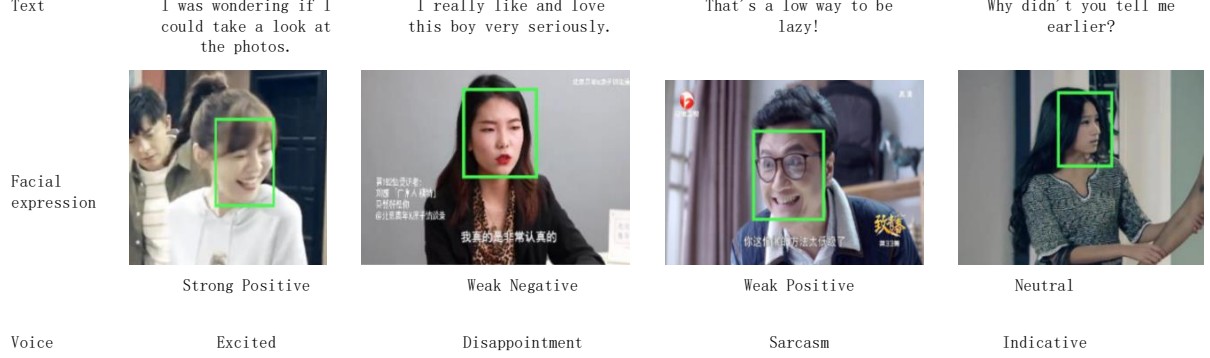

**Figure 6.** Examples in the CH-SIMS dataset. The green box in the image captures the speaker's facial expression.

CMU-MOSI [46] (Multimodal Opinion Level Sentiment Intensity) is a multimodal dataset with character subjective sentiment and sentiment intensity annotations. It contains 2199 multimodal samples from 93 YouTube videos, with each video ranging from 2–5 min and featuring 89 different speakers. Each video is annotated with sentiment intensity, ranging from strong positive to strong negative on a scale from $-3$ to 3.

Another dataset is CMU-MOSEI [21] (CMU Multimodal Opinion Sentiment and Emotion Intensity), an upgraded version of the CMU-MOSI dataset and one of the largest sentiment analysis datasets covering multiple fields, including sentiment recognition. CMU-MOSEI contains 23,453 manually annotated video clips from 5000 videos on YouTube, including 1000 different speakers and 250 different topics, covering almost all topics in daily life. CMU-MOSEI uses the same annotation method as CMU-MOSI.

In addition, considering the research on multimodal sentiment analysis in the Chinese community, we also used CH-SIMS [47], a refined Chinese multimodal dataset. It contains 2281 samples from 60 videos collected from movies, TV shows, and variety shows. Compared to the first two datasets, it not only includes multimodal sentiment labels but also provides independent fine-grained single-modality sentiment labels for each sample. Each label in this dataset is manually annotated from $-1$ (strongly positive) to 1 (strongly negative). The statistical information of these three datasets is shown in Table 2.

**Table 2.** Statistics of CMU-MOSI, CMU-MOSEI, and CH-SIMS datasets.

| Dataset | Train | Validation | Test | All |
| --- | --- | --- | --- | --- |
| CMU-MOSI | 1284 | 229 | 686 | 2199 |
| CMU-MOSEI | 16,326 | 1871 | 4659 | 22,856 |
| CH-SIMS | 1368 | 456 | 457 | 2281 |

4.1.2. Data Processing

We targeted the different modalities for processing. For the text modality, we used the BERT-based-uncased model to encode the CMU-MOSI and CMU-MOSEI datasets. In addition, for the Chinese multimodal sentiment dataset CH-SIMS, we used the BERT-based-Chinese model for text encoding. This step helps to transform text data into vector representations with rich semantic information.

When processing the speech modality, we used the COVAREP tool to extract audio features, including pitch, glottal source parameters, and 12 Mel-frequency cepstral coefficients (MFCCs). These features capture sound frequencies, voice source properties, and acoustic features in speech, providing important information for sentiment analysis. For the CH-SIMS dataset, we used the Librosa toolkit in Python to extract speech features such as log fundamental frequency, constant-Q chromatograms, and 20 MFCCs.

For visual modality, we used the Facet tool to extract 35 facial features for the CMU-MOSI and CMU-MOSEI datasets, which record facial muscle movements related to sentiment. For the Chinese sentiment dataset CH-SIMS, we used the OpenFace 2.0 toolkit to extract 17 facial action units, 68 facial landmarks, and some features related to head posture and eye movements. These facial features capture information related to facial expressions in sentiment expression, providing important visual data for multimodal sentiment analysis.

4.1.3. Baseline

In the field of multimodal sentiment analysis, there exists a series of different baseline models, each with its own characteristics. In order to comprehensively verify the performance of the method proposed in this paper, we compared it with many current methods, which mainly include the following:

TFN [24]. The tensor fusion network is a tensor-fusion-based method that computes the triple Cartesian product between three modalities to explicitly capture intramodal-

ity and intermodality dynamic information. It utilizes tensor operations to capture the interaction and fusion of multimodal information.

LMF [25]. Similar to TFN, low-rank multimodal fusion also relies on tensor operations, but it cleverly uses modality-specific low-rank factors to more efficiently compute multimodal representations, improving fusion efficiency while ensuring information quality.

MulT [30]. Multimodal Transformer adopts a bidirectional cross-modal attention mechanism to calculate the relation between two different modalities separately. The method is based on Transformer architecture, which can better capture dynamic information between different modalities.

MISA [48]. Modality-invariant and-specific representations for multimodal sentiment analysis. MISA uses a subspace learning approach to map each modality to two different subspaces for learning, providing a comprehensive view of multimodal representation learning and achieving better fusion results.

Self-MM [49]. The self-supervised multitask multimodal sentiment analysis network designs an unimodal label generation module based on self-supervised learning to obtain independent unimodal representations. It utilizes self-supervised learning to improve model performance. Also, it jointly trains multimodal and unimodal tasks to learn modal consistency and variability.

AMML [50]. Adaptive multimodal meta-learning uses a meta-learning approach to train unimodal networks and applies them to multimodal inference. This method focuses on network adaptability and optimizes unimodal representations through adaptive learning rate adjustment for better multimodal fusion.

MMIM [51]. MultiModal InfoMax proposes a hierarchical maximization of mutual information framework, which improves the consistency and information density of multimodal representations by maximizing mutual information and preserves task-relevant information through multimodal fusion.

EMT [33]. Efficient Multimodal Transformer proposes an efficient network based on the Transformer architecture for integrating multimodal information. This network utilizes unimodal encoders to obtain multimodal representations and enables mutual learning between multimodal global representations and unimodal feature sequences.

### 4.1.4. Hyper-Parameter Setting

We use the Pytorch in deep learning to build our model and optimize it with the Adam optimizer, and we adopt an early-stop strategy. Table 3 shows the parameter settings for CCDA trained on CMU-MOSI, CMU-MOSEI, and CH-SIMS datasets. In the cross-attention section, we adopt the same hyper-parameter settings as EMT, and in the self-attention section, we use the Transformer parameter settings in MulT. To reflect the accuracy of the results, we conducted five experiments and averaged each metric in the experimental results.

**Table 3.** Hyper-parameter settings of CCDA on three datasets.

| Hyper-Parameter | CMU-MOSI | CMU-MOSEI | CH-SIMS |
|---|---|---|---|
| Batch size | 32 | 16 | 32 |
| Early stop (epochs) | 16 | 8 | 16 |
| Learning rate | $1 \times 10^{-3}$ | $1 \times 10^{-4}$ | $1 \times 10^{-3}$ |
| Optimizer | Adam | Adam | Adam |
| Dimension of feature and representation | 128 | 128 | 128 |
| Transformer layers in cross-attention | 3 | 2 | 4 |
| Cross-attention heads | 4 | 4 | 4 |
| Transformer layers in self-attention | 2 | 2 | 2 |
| Attention dropout | 0.1 | 0.1 | 0.1 |
| Stacked LSTM layers for self-attention | 2 | 2 | 2 |
| Stacked LSTM dropout | 0.1 | 0.1 | 0.1 |
| $\lambda$ in cross-correlation loss | $5 \times 10^{-5}$ | $5 \times 10^{-5}$ | $1 \times 10^{-3}$ |
| Projector dims in cross-correlation loss | 1024 | 1024 | 256 |

*4.2. Result Analysis*

4.2.1. Evaluation Metrics

In regression tasks, we mainly use two metrics to measure model performance: mean absolute error (MAE) and Pearson correlation coefficient (Corr). MAE is used to measure the average absolute error between the model's predicted values and the true labels, with lower values indicating better model performance. Corr is used to measure the correlation between the model's predicted results and the true labels, with values closer to 1 indicating better model performance. Additionally, we also convert the model's output results into classification task metrics, including Acc-k and F1-score. Acc-2, Acc-5, and Acc-7 on the CMU-MOSI and CMU-MOSEI datasets and Acc-2, Acc-3, and Acc-5 on CH-SIMS are used to evaluate the model's accuracy in multiclassification tasks, with larger values indicating better model performance. F1-score represents the harmonic mean of precision and recall and is used to evaluate the balance between positive and negative categories. A higher F1-score indicates better model performance in classification tasks.

4.2.2. Quantitative Analysis

The experimental data for TFN, LMF, MulT, MISA, Self-MM, and MMIM come from [51]. For the other models, we conducted five experiments on each of the three datasets using publicly available source code and averaged the experimental results for each model. In all evaluation metrics, except for MAE, larger values indicate better model performance. The experimental results are compared in Tables 4–6.

Table 4 shows the model's results on the CMU-MOSI dataset. Compared to the EMT model, CCDA improved by 0.009 on the regression metrics MAE and Corr. In terms of classification task metrics, CCDA improved by 0.6% on Acc-2 and Acc-5 and 0.7% on Acc-7 and achieved a 0.6% improvement in F1-score over the best model. Similarly, as shown in Table 5, CCDA's performance on CMU-MOSEI improved by 0.003 on MAE, 0.006 on Corr, 0.5% on Acc-7, 0.4% on Acc-5, 0.6% on Acc-2, and 0.7% on F1-score compared to EMT. Table 6 shows the experimental results of the model on CH-SMIS, where CCDA achieved better results on some metrics, such as 0.006 on MAE, 0.005 on Corr, 1.4% on Acc-3, 1.2% on Acc-2, and 0.9% on F1-score. However, its performance on the 5-classification task was slightly worse than that of the EMT model. We believe that while CCDA improves coarse-grained sentiment classification, it does not improve much for fine-grained classification.

The experimental results show that the CCDA model using cross-modal attention and self-attention is able to learn intra- and intermodal dynamics. Dual-attention makes the model analyze the sample more comprehensively, and the cross-correlation loss enables some degree of interaction between different levels of dual-attention mechanism. In addition, the relevant coefficients guide the multimodal feature fusion stage, which allows the model to improve performance while increasing the model's generalization ability.

**Table 4.** Experiments on CMU-MOSI. Where ↑ indicates that the higher the metric the stronger the performance of the model, and ↓ is the opposite. Bold numbers indicate the model with the best results at that metric.

| Models | MAE (↓) | Corr (↑) | Acc-7 (↑) | Acc-5 (↑) | Acc-2 (↑) | F1 (↑) |
|---|---|---|---|---|---|---|
| TFN [24] | 0.901 | 0.698 | 34.9 | - | 80.8 | 80.7 |
| LMF [25] | 0.917 | 0.695 | 33.2 | - | 82.5 | 82.4 |
| MulT [30] | 0.846 | 0.725 | 40.4 | 46.7 | 83.4 | 83.5 |
| MISA [48] | 0.804 | 0.764 | - | - | 82.1 | 82.0 |
| Self-MM [49] | 0.717 | 0.793 | 46.4 | 52.8 | 84.6 | 84.6 |
| MMIM [51] | 0.712 | 0.790 | 46.9 | 53.0 | 85.3 | 85.4 |
| AMML [50] | 0.723 | 0.792 | 46.3 | - | 84.9 | 84.8 |
| EMT [33] | 0.705 | 0.798 | 47.4 | 54.1 | 85.0 | 85.0 |
| Ours | **0.696** | **0.807** | **48.0** | **54.8** | **85.7** | **85.6** |

**Table 5.** Experiments on CMU-MOSEI. Where ↑ indicates that the higher the metric the stronger the performance of the model, and ↓ is the opposite. Bold numbers indicate the model with the best results at that metric.

| Models | MAE (↓) | Corr (↑) | Acc-7 (↑) | Acc-5 (↑) | Acc-2 (↑) | F1 (↑) |
|---|---|---|---|---|---|---|
| TFN [24] | 0.593 | 0.700 | 50.2 | - | 82.5 | 82.1 |
| LMF [25] | 0.623 | 0.677 | 48.0 | - | 82.0 | 82.1 |
| MulT [30] | 0.564 | 0.731 | 52.6 | 54.1 | 83.5 | 83.6 |
| MISA [48] | 0.568 | 0.724 | - | - | 84.2 | 84.0 |
| Self-MM [49] | 0.533 | 0.766 | 53.6 | 55.4 | 85.0 | 85.0 |
| MMIM [51] | 0.536 | 0.764 | 53.2 | 55.0 | 85.0 | 85.1 |
| AMML [50] | 0.614 | 0.776 | 52.4 | - | 85.3 | 85.2 |
| EMT [33] | 0.527 | 0.774 | 54.5 | 56.3 | 86.0 | 86.0 |
| Ours | **0.524** | **0.780** | **55.0** | **56.7** | **86.6** | **86.7** |

**Table 6.** Experiments on CH-SIMS. Where ↑ indicates that the higher the metric the stronger the performance of the model, and ↓ is the opposite. Bold numbers indicate the model with the best results at that metric.

| Models | MAE (↓) | Corr (↑) | Acc-5 (↑) | Acc-3 (↑) | Acc-2 (↑) | F1 (↑) |
|---|---|---|---|---|---|---|
| TFN [24] | 0.437 | 0.582 | - | - | 77.1 | 76.9 |
| LMF [25] | 0.438 | 0.578 | - | - | 77.4 | 77.4 |
| MulT [30] | 0.442 | 0.581 | 40.0 | 65.7 | 78.2 | 78.5 |
| MISA [48] | 0.447 | 0.563 | - | - | 76.5 | 76.6 |
| Self-MM [49] | 0.411 | 0.601 | 43.1 | 66.1 | 78.6 | 78.6 |
| MMIM [51] | 0.422 | 0.597 | 42.0 | 65.5 | 78.3 | 78.2 |
| AMML [50] | 0.437 | 0.583 | 41.2 | 64.2 | 78.0 | 78.1 |
| EMT [33] | 0.396 | 0.623 | **43.5** | 67.4 | 80.1 | 80.1 |
| Ours | **0.393** | **0.628** | 43.3 | **68.3** | **81.1** | **81.0** |

### 4.3. Ablation Study

To validate the role of the dual-attention mechanism in the CCDA model and the effects of the multimodal fusion strategy and cross-correlation loss on the performance of the model, we conducted ablation experiments on two datasets, CMU-MOSI and CH-SIMS.

#### 4.3.1. Dual-Attention Mechanisms

The MulT model first uses multiple cross-modal attention mechanisms between the bimodal features and later uses a Transformer encoder. Throughout the training process, the model does not capture modality-specific intramodal information, but, rather, directly interacts cross-modally. While this enables unimodal features to perceive affective information from neighboring modalities upfront, this will lose the modality specific information. EMT splices the unimodal representation as a global representation and selects the Transformer encoder to interact with the global and unimodal representations, but in the process does not model each modality individually, which can result in the model failing to capture affective information that exists within a single modality.

We designed a set of experiments to verify the effect of different mechanisms in dual-attention on model performance, as shown in Tables 7 and 8. The first rows of Tables 7 and 8 validate the model performance in the case of using only the unimodal self-attention mechanism, where we used the relevant coefficient to guide the unimodal representations. The final multimodal feature has only three unimodal representations and does not contain the global representation in standard CCDA. The second row verifies the model performance in the case where only the cross-modal attention mechanism is used, in which case the multimodal features are global representations, not containing unimodal representations, and the relevant coefficients cannot be used. Neither of these cases uses the cross-correlation loss. The third row indicates that we use the standard CCDA model for training.

**Table 7.** Impact of dual-attention in CCDA on CMU-MOSI. Where ↑ indicates that the higher the metric the stronger the performance of the model, and ↓ is the opposite. Bold numbers indicate the model with the best results at that metric.

|  | MAE (↓) | Corr (↑) | Acc-7 (↑) | Acc-5 (↑) | Acc-2 (↑) | F1 (↑) |
|---|---|---|---|---|---|---|
| Only self-attention | 0.734 | 0.764 | 45.1 | 51.9 | 83.0 | 83.0 |
| Only cross-attention | 0.722 | 0.787 | 46.4 | 53.2 | 84.4 | 84.5 |
| Standard CCDA | **0.696** | **0.807** | **48.0** | **54.8** | **85.7** | **85.6** |

**Table 8.** Impact of dual-attention in CCDA on CH-SIMS. Where ↑ indicates that the higher the metric the stronger the performance of the model, and ↓ is the opposite. Bold numbers indicate the model with the best results at that metric.

|  | MAE (↓) | Corr (↑) | Acc-5 (↑) | Acc-3 (↑) | Acc-2 (↑) | F1 (↑) |
|---|---|---|---|---|---|---|
| Only self-attention | 0.443 | 0.602 | 40.7 | 65.7 | 78.4 | 78.3 |
| Only cross-attention | 0.415 | 0.613 | 41.9 | 66.8 | 79.9 | 79.9 |
| Standard CCDA | **0.393** | **0.628** | **43.3** | **68.3** | **81.1** | **81.0** |

The data in the table show that when using self-attention, the model is unable to focus on cross-modal interaction information and only fuses the representations of each modality at a later stage. While the model performance improves when using only cross-attention, this is due to the fact that it discriminates the sentiment attributes of the sample as a whole from a global perspective, and compared to self-attention, cross-attention tends to select the information that is the most beneficial to the overall judgment when performing interactions. In the standard CCDA model, the model's performance is optimal when dual-attention is used at the same time, which suggests that CCDA retains as much of the affective information in dual-attention as possible.

### 4.3.2. Fusion Strategy with Relevant Coefficients

Before performing multimodal fusion in the model, we adjusted the unimodal representations based on the relevant coefficients computed between unimodal representations and their respective initial modality representations. Subsequently, these representations were concatenated with the global multimodal representation. To validate the effectiveness of our proposed fusion strategy, we conducted experiments on both Chinese and English datasets. We compared the performance of models with and without considering unimodal relevant coefficients, where the unimodal representations, computed after self-attention and subsequent Bi-LSTMs, were directly concatenated with the global multimodal representation, and then fed into the fusion and prediction module. We also compared these results with the standard version of CCDA. The comparative experimental results are shown in Tables 9 and 10.

According to Tables 9 and 10, it is evident that in multimodal fusion, the model's performance significantly improves when unimodal features are augmented with relevant coefficients compared to direct concatenation. Specifically, there is a 1.5% improvement in Acc-7. Therefore, the use of relevant coefficients in the multimodal feature fusion stage enables the model to analyze the relations between the self-attention modality representations and the source feature representations, and, thus, to achieve higher accuracy on multiclassification.

**Table 9.** Impact of correlation coefficients in fusion strategy on CMU-MOSI. Where ↑ indicates that the higher the metric the stronger the performance of the model, and ↓ is the opposite. Bold numbers indicate the model with the best results at that metric.

|  | MAE (↓) | Corr (↑) | Acc-7 (↑) | Acc-5 (↑) | Acc-2 (↑) | F1 (↑) |
|---|---|---|---|---|---|---|
| Direct Concat | 0.713 | 0.790 | 46.5 | 53.8 | 85.2 | 85.2 |
| Standard CCDA | **0.696** | **0.807** | **48.0** | **54.8** | **85.7** | **85.6** |

**Table 10.** Impact of correlation coefficients in fusion strategy on CH-SIMS. Where ↑ indicates that the higher the metric the stronger the performance of the model, and ↓ is the opposite. Bold numbers indicate the model with the best results at that metric.

|  | MAE (↓) | Corr (↑) | Acc-5 (↑) | Acc-3 (↑) | Acc-2 (↑) | F1 (↑) |
|---|---|---|---|---|---|---|
| Direct Concat | 0.408 | 0.614 | 41.2 | 66.4 | 80.4 | 80.4 |
| Standard CCDA | **0.393** | **0.628** | **43.3** | **68.3** | **81.1** | **81.0** |

### 4.3.3. Cross-Correlation Loss

Additionally, this study assumes a certain degree of cross-correlation between self-attention and cross-attention. Thus, we introduced a cross-correlation loss function to facilitate indirect interaction between these two attention mechanisms. To assess the impact of cross-correlation loss on model performance, we conducted ablation experiments on the CMU-MOSI and CH-SIMS datasets, as shown in Tables 11 and 12.

**Table 11.** Impact of cross-correlation loss in the objective function on CMU-MOSI. Where ↑ indicates that the higher the metric the stronger the performance of the model, and ↓ is the opposite. Bold numbers indicate the model with the best results at that metric.

|  | MAE (↓) | Corr (↑) | Acc-7 (↑) | Acc-5 (↑) | Acc-2 (↑) | F1 (↑) |
|---|---|---|---|---|---|---|
| w/o corr loss | 0.708 | 0.795 | 47.4 | 54.2 | 84.9 | 84.9 |
| Standard CCDA | **0.696** | **0.807** | **48.0** | **54.8** | **85.7** | **85.6** |

**Table 12.** Impact of cross-correlation loss in the objective function on CH-SIMS. Where ↑ indicates that the higher the metric the stronger the performance of the model, and ↓ is the opposite. Bold numbers indicate the model with the best results at that metric.

|  | MAE (↓) | Corr (↑) | Acc-5 (↑) | Acc-3 (↑) | Acc-2 (↑) | F1 (↑) |
|---|---|---|---|---|---|---|
| w/o corr loss | 0.400 | 0.610 | 42.0 | 66.7 | 80.1 | 80.1 |
| Standard CCDA | **0.393** | **0.628** | **43.3** | **68.3** | **81.1** | **81.0** |

It can be observed that adding cross-correlation loss to the objective function significantly enhances the model's performance. This improvement is particularly pronounced in multiclass tasks, indicating that cross-correlation loss has a substantial impact on model performance in multimodal sentiment analysis. Further analysis reveals that cross-correlation loss establishes a closer connection between self-attention and cross-attention in the model, enabling better integration of information from multimodal data. This indirect interaction helps the model better understand the relationships between different modalities, thereby improving overall sentiment analysis performance. In multimodal sentiment analysis tasks, such enhanced connectivity is highly beneficial. Moreover, the results on different datasets demonstrate the universality of the improvement brought by cross-correlation loss, indicating that it is not limited to specific datasets. This strengthens the scalability and generality of our approach.

### 4.3.4. Scaling Factor in Cross-Correlation Loss

When calculating the cross-correlation loss, the model expands the dimensions of the feature sequences. As a result, the values of elements in the correlation matrix become relatively large. To balance the cross-correlation loss in the objective function, we introduced scaling factors. Figure 7 illustrates the impact of scaling factors on the final results. Since we set different feature dimensions for unimodal features from different datasets (128 for CMU-MOSI and CMU-MOSEI, 32 for CH-SIMS), and applied different linear mapping layers for dimension expansion when calculating the cross-correlation loss for different datasets, the optimal scaling factors also vary. Specifically, we used $5 \times 10^{-5}$ for CMU-MOSI and $1 \times 10^{-3}$ for CH-SIMS.

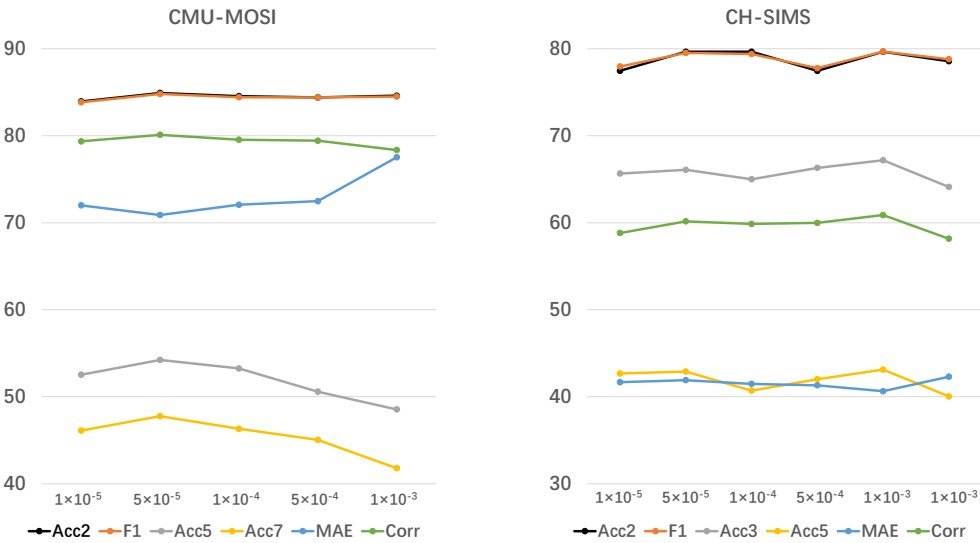

**Figure 7.** Impact of scaling weights in cross-correlation loss. Where a lower MAE (blue line) indicates better model performance, showing an opposite trend to the other metrics.

## 5. Conclusions

In this paper, we introduced the cross-correlation in dual-attention (CCDA) model aimed at fusing multimodal features and perceiving human sentiment analysis. We used dual-attention to obtain information about the intra- and intermodal dynamics contained in the samples from different perspectives, and in order to capture the relation that exists between different attention mechanisms, we propose the cross-correlation loss, which allows the cross-modal attention and the self-attention mechanism to complete a nondirective interaction. In addition, we introduce a new fusion strategy in the multimodal feature fusion stage by using correlation coefficients, which allows the initial unimodal representation to guide the multimodal fusion.

We conducted comprehensive experiments on three commonly used public datasets in the multimodal sentiment analysis domain, including CMU-MOSI, CMU-MOSEI, and CH-SIMS. We compared the CCDA model with baseline models and found that our model demonstrated a significant advantage on all three datasets. Through experimentation, we demonstrated the strong performance of the CCDA model in multimodal sentiment analysis tasks, offering new insights for further research and applications in this field.

Since the Transformer is used in this study, an issue that cannot be ignored is the number of parameters of the model, which increases rapidly as the number of attention block increases. In addition, the cross-correlation loss as well as the relevant coefficients in this study were calculated using matrix multiplication, which increases the computational complexity of the model, and there is still some redundant information in the calculation process.

Future research work:In view of the problems encountered in this study, future research efforts should focus on (1) reducing the number of parameters of the model while ensuring the model performance, (2) reducing the computational complexity of the model, and (3) further reducing the redundant information generated during the training process of the model.

Given the challenges faced in real-world multimodal sentiment analysis, especially in scenarios involving missing modal information, future research could focus on enhancing the model's robustness and accuracy in handling missing modal information. This would ensure the effectiveness and reliability of the model in a wider range of practical applications.

**Author Contributions:** Conceptualization, P.W.; methodology, P.W.; software, P.W.; validation, P.W., S.L., and J.C.; formal analysis, P.W.; investigation, P.W.; resources, P.W.; data curation, P.W.; writing—original draft preparation, P.W.; writing—review and editing, P.W.; visualization, P.W.; supervision, P.W.; project administration, P.W.; funding acquisition, S.L. All authors have read and agreed to the published version of the manuscript.

**Funding:** This work was supported by the National Natural Science Foundation of China (61762085) and the Natural Science Foundation of Xinjiang Uygur Autonomous Region Project (2019D01C081).

**Institutional Review Board Statement:** Not applicable.

**Informed Consent Statement:** Not applicable.

**Data Availability Statement:** The data presented in this study are available in https://github.com/CMU-MultiComp-Lab/CMU-MultimodalSDK, reference number [21,46] and https://github.com/thuiar/MMSA, reference number [47]. These data were derived from the following resources available in the public domain: https://drive.google.com/drive/folders/1A2S4pqCHryGmiqnNSPLv7rEg63WvjCSk.

**Conflicts of Interest:** The authors declare no conflicts of interest.

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
