# Peer review of "CCDA: A Novel Method to Explore the Cross-Correlation in Dual-Attention for Multimodal Sentiment Analysis"

_applsci, doi:10.3390/app14051934_

Round 1

Reviewer 1 Report

Comments and Suggestions for Authors
  1. Originality and Contribution: The paper introduces an innovative approach in multimodal sentiment analysis, which is commendable. The CCDA model's use of dual-level attention mechanisms and a cross-correlation loss function is a notable contribution.
  2. Experimental Results: The experiments conducted are thorough, and the results demonstrate the model's effectiveness. However, it would be beneficial to include a more detailed analysis of why the model outperforms existing methods.
  3. Future Work: The discussion on future enhancements is insightful. It would be enriching to see more specific suggestions or potential directions for future research.

    Main Research Question: Clarify the primary research question for a stronger focus.

    Originality and Relevance: Highlight how the CCDA model uniquely addresses gaps in integrating multiple data types for sentiment analysis.

    Contribution Compared to Existing Literature: Include a comparative analysis with existing models, emphasizing specific improvements or advancements.

    Methodological Improvements: Suggest testing the model with different datasets or control groups to validate its robustness. Exploring performance with skewed or unbalanced data could also provide valuable insights.

    Consistency of Conclusions with Evidence: Ensure the conclusions align with the experimental results, and address whether all aspects of the dual-level attention mechanisms contribute equally to the model's performance.

    References Appropriateness: Broaden the range of references to include recent, challenging, or alternative perspectives on multimodal sentiment analysis.

    Additional Comments on Data and Discussion: Enhance the discussion on anomalies or unexpected findings in the data. Expand the contextualization within the field in the discussion section.

In conclusion, the paper is a valuable addition to the field of multimodal sentiment analysis.

Author Response

We have clarified the main research questions and strengthened the theoretical underpinnings. The experimental results were analyzed and a set of ablation experiments were added to verify the reliability of the study, and finally we added a discussion of future research work.

Reviewer 2 Report

Comments and Suggestions for Authors

The study presents a cross-correlation in dual-attention method for multimodal sentiment analysis. Although the topic is interesting, there are several shortcomings in this manuscript.

Throughout the manuscript, statements are not being supported by references. The authors simply list 3-9 references in simple sentences/statements or even words which is not a good or ethical practice. Each reference should be used to support the claims of the study and provide a better understanding of the background.

The need for this study is not clear. The authors do not present other related studies and their results. They simply name other methods that have been used. A thorough and detailed synthesis of other related studies should be conducted in order to highlight the current literature gap.

There is also a lack of a detailed result analysis which goes in more detail regarding the differences of the methods adopted.

A serious issue in this study is the lack of a discussion section. The authors should integrate a detailed discussion in which they further expand upon their results while making connections and comparisons to those of other related studies. Not presenting other works and comparing the results greatly affects the impact of the study.

The conclusion section does not provide any conclusive remarks but simply restate what has already been mentioned in the text. The authors should highlight the main findings that arose as well as the implications of their method. Additionally, the authors should clearly mention the limitations of their study and method and provide guidelines for future research.

Comments on the Quality of English Language

The quality of English is satisfactory. Only some minor corrections are required.

Author Response

We revise the related work partly and discuss the theoretical foundations of this study at the end of this section. The experimental results are analyzed and the concluding section discusses the methodology used in the study and identifies research directions for future work.

Reviewer 3 Report

Comments and Suggestions for Authors

Let’s us express my interest in your recent development in attention mechanisms. I'm find quite interesting the idea apart from using self-attention mechanism, also a cross-attention. Because of widespread usage of multimodal datasets, it is necessary to find the alignment of some corresponding elements of diferent modalities in the same time. This allows to find the dependency of each other and propose plausible hypothesis on data and usage of different modalities.

Overall, the article has no issues, apart from table on page 3. I suggest to fit the table to width of the page, because in general it is a good style guideline to have images and tables to fit the margins.

Now to overall impressions part. The paper is well written, has enough formulae, tables, figures and necessary explanations, as well as experiments. 

Since I have no questions about this particular article, I leave some suggestions.

1. The first and main, more in depth explanation of cross attention mechanism and its relation to correlation, self of multihead attention, with respect to image of attention/correlation matrices. It is necessary to portray the image of such matrices “in the wild”.

2. Second, it is necessary to portray the event which correlates within the multimodal data. For instance, you study a dataset of some utterances, where people express some opinions about something. Imagine person expresses emotion of regret, disdain or disbelief after or while saying term or word “unfair”. It means that the speech and emotional channels do present some cross-attention between some events, hence you have key-value pair of speech “unfair” and corresponding pair “negative emotion”.

3. Since you study temporal features, it would be interesting to study how cross-attention works in conjunction with temporal features, i.e. when events are occuring in near time, but some delay is present. 

4. Lastly, speaking about some data examples. Maybe, if speaking about images, or schemes of multimodal recognition structure it would be good to add some sample images of the multimodal dataset, that underscore different nature of the studied data but aslo visual representation of the data, simply saying how one modality relates to another. 

In overall I have a very good impression about your paper and recommend its publishing after fixing a sole issue with table 1. It is well written and there is no need of adding more edits apart from this one. I want to underscore that you got good overall results in the experimental section, which allowed to achieve better performance, compared to the known results, that were not using your approach to the data.

Author Response

We added a heat map of the cross-correlation matrix and showed sample examples from the dataset.

Round 2

Reviewer 2 Report

Comments and Suggestions for Authors

The authors have made extensive changes and additions to their manuscript which I believe have improved its quality.

The only issue that still remains is that the authors still “simply list 3-9 references in simple sentences/statements or even words which is not a good or ethical practice”. There hasn’t been any change in this regard and it occurs in several places including the introduction where it is more evident.

Comments on the Quality of English Language

The quality of English has improved. Some minor issues that existed earlier have been corrected.

Author Response

We revised the section of the introduction that describes the work related to multimodal sentiment analysis.
